# Effect of Early Supraglottic Airway Device Insertion on Chest Compression Fraction during Simulated Out-of-Hospital Cardiac Arrest: Randomised Controlled Trial

**DOI:** 10.3390/jcm11010217

**Published:** 2021-12-31

**Authors:** Loric Stuby, Laurent Jampen, Julien Sierro, Maxime Bergeron, Erik Paus, Thierry Spichiger, Laurent Suppan, David Thurre

**Affiliations:** 1Genève TEAM Ambulances, Emergency Medical Services, CH-1201 Geneva, Switzerland; d.thurre@gt-ambulances.ch; 2ESAMB—École Supérieure de Soins Ambulanciers, College of Higher Education in Ambulance Care, CH-1231 Conches, Switzerland; laurent.jampen@edu.ge.ch; 3Compagnie d’Ambulances de l’Hôpital du Valais, Emergency Medical Services, CH-1920 Martigny, Switzerland; julien.sierro@hopitalvs.ch; 4Société de Transport en Ambulance Régionale, STAR Ambulances, Emergency Medical Services, CH-1066 Épalinges, Switzerland; maximeb@starambulances.ch; 5SPSL—Service de Protection et Sauvetage Lausanne, Emergency Medical Services, CH-1005 Lausanne, Switzerland; Erik.Paus@spsl-lausanne.ch; 6ES ASUR, Vocational Training College for Registered Paramedics and Emergency Care, CH-1052 Le Mont-sur-Lausanne, Switzerland; t.spichiger@es-asur.ch; 7Ambulance Riviera, Association Sécurité Riviera, Emergency Medical Services, CH-1814 La Tour-de-Peilz, Switzerland; 8Division of Emergency Medicine, Department of Anaesthesiology, Clinical Pharmacology, Intensive Care and Emergency Medicine, Geneva University Hospitals and Faculty of Medicine, CH-1211 Geneva, Switzerland; laurent.suppan@hcuge.ch

**Keywords:** Emergency Medical Services, paramedics, airway, Supraglottic Airway Device, Cardiac Arrest, i-gel^®^, CPR, prehospital, resuscitation, Chest Compression Fraction

## Abstract

Early insertion of a supraglottic airway (SGA) device could improve chest compression fraction by allowing providers to perform continuous chest compressions or by shortening the interruptions needed to deliver ventilations. SGA devices do not require the same expertise as endotracheal intubation. This study aimed to determine whether the immediate insertion of an i-gel^®^ while providing continuous chest compressions with asynchronous ventilations could generate higher CCFs than the standard 30:2 approach using a face-mask in a simulation of out-of-hospital cardiac arrest. A multicentre, parallel, randomised, superiority, simulation study was carried out. The primary outcome was the difference in CCF during the first two minutes of resuscitation. Overall and per-cycle CCF quality of compressions and ventilations parameters were also compared. Among thirteen teams of two participants, the early insertion of an i-gel^®^ resulted in higher CCFs during the first two minutes (89.0% vs. 83.6%, *p* = 0.001). Overall and per-cycle CCF were consistently higher in the i-gel^®^ group, even after the 30:2 alternation had been resumed. In the i-gel^®^ group, ventilation parameters were enhanced, but compressions were significantly shallower (4.6 cm vs. 5.2 cm, *p* = 0.007). This latter issue must be addressed before clinical trials can be considered.

## 1. Introduction

### 1.1. Background

High-quality chest compressions are mandatory to increase the probability of survival after out-of-hospital cardiac arrest (OHCA) [1,2]. One of the key determinants of survival and a favourable neurological outcome is the chest compression fraction (CCF), i.e., the proportion of time spent performing compressions during cardiopulmonary resuscitation (CPR) [3,4,5,6,7,8,9,10]. An increase of 10% in CCF can even yield an 11% increase in survival [6].

The optimal airway management strategy for OHCA is still debated [11,12,13,14,15,16,17], and airway management manoeuvres might lead to interruptions in chest compressions, thereby preventing rescuers from achieving high CCFs. Even though endotracheal intubation (ETI) is often considered as the gold standard, this technique requires great skill that must be maintained by regular practice to be both safe and efficient [18,19,20]. However, most prehospital providers are either insufficiently trained in ETI or lack the possibility of maintaining this skill. Supraglottic airway (SGA) devices might therefore represent an adequate solution in many settings, as their use does not require the same level of expertise. There are many different SGA devices, some of which are easier to use than others. Converse to traditional laryngeal mask airways, the i-gel^®^ (Intersurgical Ltd., Wokingham, UK) embeds a thermoplastic cuff that does not need to be inflated, making its insertion quicker and allowing high success rates to be achieved [21,22,23,24,25,26,27,28,29,30,31]. In addition, SGA devices seem to confer a certain degree of airway protection, as the rates of regurgitation and aspiration do not significantly differ from those seen with ETI in case of OHCA [32] and are much lower than those reported when a bag-valve-mask (BVM) device is used [33,34]. Moreover, the use of the i-gel^®^ is not associated with significant leaks during the delivery of continuous chest compressions [35]. This ensures a high reliability of end-tidal CO2 measurements, an important feature, as capnography is increasingly used to monitor and optimise CPR quality and to detect ROSC [1].

Our hypothesis was that, in case of OHCA, the early insertion of an i-gel^®^ device without prior BVM ventilation could improve CCF while allowing a similar time than the standard approach to elapse before the first effective ventilation could be delivered.

### 1.2. Objectives

The primary aim of this study was to determine whether the immediate insertion of an i-gel^®^ while providing continuous chest compressions, followed by asynchronous ventilations (experimental approach) during the two minutes following the first rhythm analysis allows for the generation of higher CCFs than the 30 compressions: 2 ventilations (standard) approach in a simulated model of OHCA.

The secondary objective was to compare the impact of this approach on CPR quality and ventilation parameters [1,36,37].

## 2. Materials and Methods

### 2.1. Study Design

This was a multicentre, parallel, randomised, superiority, simulation study designed in accordance with the Standard Protocol Items: Recommendations for Interventional Trials (SPIRIT) statement [38], the detailed protocol of which was previously published [39]. Results are reported according to the Consolidated Standards of Reporting Trials (CONSORT) statement (Appendix A) [40]. The trial was prospectively registered: NCT04736446 (3 February 2021).

### 2.2. Setting

In Switzerland, paramedics follow a 3-year training curriculum and are trained to treat most prehospital emergencies autonomously. They can administer several Advanced Life-Support (ALS) treatments according to emergency care protocols validated by prehospital medical directors. Emergency Medical Technicians (EMTs) graduate after a 1-year curriculum and are trained to assist paramedics. The ALS teams can either be composed of two paramedics or of a paramedic teaming with an EMT. Even though these aspects are standardised, the Swiss federal system generates considerable inter-cantonal heterogeneity between Emergency Medical Services (EMS). In some regions, EMS operating in the same areas have different treatment protocols, as each service has its own medical director who can almost singlehandedly decide upon the extent of delegations. This study was performed in four different EMS where paramedics applied the standard 30:2 approach and did not routinely use the i-gel^®^ in case of OHCA.

### 2.3. Participants and Recruitment

All registered paramedics and EMTs actively working in either of these EMS were eligible for inclusion. Study team members and participants who did not attend the training path were excluded. Participants were told that the study was about OHCA management but were not informed of its specific objectives. The study was conducted on the premises of each EMS.

### 2.4. Study Sequence

The study sequence is summarised in Figure 1.

#### 2.4.1. Randomisation

Randomisation was two-tiered. First, to ensure that there would be at least one paramedic per team, an online balanced team generator was used [41]. Intra-cluster randomisation with a 1:1 ratio was then performed using an online randomiser, with each trial center representing a cluster [42]. Opaque, sealed envelopes were used to ensure the concealment of allocation.

#### 2.4.2. Standardised Workshop

The use of the i-gel^®^ device was taught to each team separately by one of the investigators (Loric Stuby, LSt) in accordance with Peyton’s approach [43,44,45], based on a standard operating procedure (Appendix A). Twenty minutes were dedicated to this workshop. This took into account the time required to gather the participants’ consent and to allow them to complete a short demographic questionnaire.

#### 2.4.3. Training Session of Experimental Approach

Following the workshop, each team was allowed 20 min of self-training supported by a custom-made demonstration video (https://cpr2-intro.swiss-cpr-studies.ch/, accessed on 30 November 2021). During this training session, an experimental approach developed by David Thurre (DT) and LSt was practised. This approach consists of the immediate placement of an i-gel^®^ device by one team member immediately after the first rhythm analysis, without prior BVM ventilation. Meanwhile, the second team member provides continuous chest compressions. Ventilations are given asynchronously at a rate of 10 per minute once the SGA device is in place [1,2]. After two minutes of continuous chest compressions, 30:2 CPR is provided “over-the-head” by one rescuer while the other performs ALS actions.

#### 2.4.4. Study Scenario

Another custom-made video, which presented the manikin’s and defibrillator’s characteristics, was displayed to the participants upon entering the study room (https://cpr2-briefing.swiss-cpr-studies.ch/, accessed on 30 November 2021). The scenario was then presented in exactly the same way to all participants regardless of the study site: “This is Michael, a 50-year-old man who suddenly collapsed 10 min ago. He is now unconscious, pale, and does not seem to be breathing. Medical reinforcement is already underway and will arrive in about ten minutes. No first responder has been dispatched by the emergency medical call center, and there is no bystander nearby”. Only then could the leader open the opaque, sealed envelope containing the specific airway management strategy the team has to apply. There was no further interaction between participants and the study team until the scenario was stopped. The simulated patient was apneic and in refractory ventricular fibrillation (VF) regardless of the number of defibrillation attempts. The scenario was stopped at T0 (first compression) + 10 min.

At the end of the scenario, participants were asked to fill a last questionnaire designed to assess their satisfaction regarding the approach they had applied and the cognitive load they had perceived during the study scenario. Participants were asked to withhold information regarding the aim and course of the study until data collection had been completed in all four EMS.

### 2.5. Equipment

Apart from the multiparametric monitor/defibrillator, which was provided with the simulation manikin (Laerdal SimMan 3G, Laerdal Medical, Stavanger, Norway), all teams had access to their usual resuscitation equipment. A size 4 i-gel^®^ device (Intersurgical Ltd., Wokingham, UK) and a lubricant recommended by the manikin’s manufacturer were added to this equipment.

### 2.6. Study Outcomes

#### 2.6.1. Primary Outcome

The primary outcome was the CCF achieved during the first two minutes of CPR (starting from first compression).

#### 2.6.2. Secondary Outcomes

Secondary outcomes were: overall (entire 10-min scenario) and per-cycle CCF (at 2–4, 4–6, 6–8 and 8–10 min); depth of compressions; proportion of compressions within (5 to 6 cm), below (<5 cm) and above (>6 cm) target value; compression rate; proportion of compressions within (100–120 compressions per minute—cpm), below (<100 cpm) and above target values (>120 cpm); proportion of compressions with complete chest recoil (<5 mm deviation from the reference value); time to first shock; time to first effective ventilation (defined as > 300 mL [46,47,48,49,50]); proportions of ventilations within, (300–700 mL), below (<300 mL) and above (>700 mL) target values; number of ventilations delivered; minute ventilation volume; provider satisfaction; and self-assessed cognitive load.

### 2.7. Data Collection, Extraction and Curation

Most data were automatically collected through the manikin’s sensors and extracted to a comma-separated values (CSV) file, thereby preventing assessment bias. Variables of interest were automatically generated using a custom-coded PHP script [51]. Data gathered on paper questionnaires were entered in duplicate using EpiData (The EpiData Association, att. Jens Lauritsen, Enghavevej 34, DK5230 Odense M, Denmark, Europe) [52], and any discrepancy was listed and resolved to minimise copying and typing errors. Missing data were treated as such. No imputation technique was used. The minute ventilation volume was calculated by dividing the total volume delivered by the ventilation time (corresponding to 600 s scenario time minus the time to first ventilation). All data that could have allowed the data analyst (DT) to identify group allocation were removed (e.g., number of insertion attempts). The groups were renamed “Teysachaux” and “Moléson”, and the curated databases were sent in a Stata DTA file format for blinded statistical analyses. All investigators were able to access the curated and coded data sets [53].

### 2.8. Sample Size Calculation

Few data were available to help with the sample size calculation. We estimated that the mean time needed to deliver two ventilations would amount to approximately 4 s, thereby adding up to 20 s per cycle. We also estimated that the initial rhythm analysis should take about 8 s, thus increasing the no-compression time to around 28 s out of the first 120 s cycle (23% of no-flow, worth 77% of CCF in the control group). In the experimental group, we estimated that the no-flow should be lower, as ventilations were to be provided without interrupting compressions (8 s only for rhythm analysis: 7% of no-flow, worth 93% of CCF). Based on observational data from pilot tests and case reviews, data variability was estimated with a standard deviation of 12. Twenty-six teams were therefore required to have a 90% chance of detecting, at the 5% significance level, a difference in CCF from 77% in the control group to 93% in the experimental group. In the original study protocol [39], the sample size calculation resulted in 24 teams because it was calculated with an online calculator [54] rather than with Stata.

### 2.9. Statistical Analysis

Variables were described using either mean (SD or 95% CI) or median [Q1; Q3] according to normality. A Student’s t-test or the Mann–Whitney U test were used accordingly. Proportions were reported with their 95% CI. Cognitive load was treated as a continuous variable. User satisfaction was assessed graphically, then tested using Fisher’s exact test. There was a minor protocol deviation, as we had initially planned to dichotomise this variable. To avoid losing valuable information, it was, however, decided not to proceed with dichotomisation. This decision was taken after protocol publication but before completion of data collection and statistical analyses. A two-sided *p*-value lower than 0.05 was considered significant. All statistical analyses were performed using Stata V15.1 (StataCorp. 2017. Stata Statistical Software: Release 15. College Station, TX, USA, StataCorp LLC).

## 3. Results

Fifty-two participants were recruited (44 paramedics and 8 EMTs) between March and May 2021. They were divided into 26 teams allocated equally to both groups (Figure 2). Their characteristics are detailed in Table 1. All were analysed by original assigned groups.

In the experimental group, the successful insertion rate was of 84.6% (11/13) at first try and of 100% at the second one. Among this group, 11/13 teams switched to 30:2 CPR, as expected (2 min after the first rhythm analysis). The CCF was consistently higher when the experimental approach was used (Table 2).

Applying the experimental approach led to shallower compressions with a difference higher than 0.6 cm (median (Q1; Q3) 4.6 cm (4.3; 5.0) versus 5.2 cm (4.9; 5.3), *p* = 0.007). Accordingly, the mean proportion of compressions within the depth target (5 to 6 cm) was lower in the experimental group (41.7% (95% CI 28.2–55.3) versus 66.5% (95% CI 51.5–81.4), *p* = 0.01). All out-of-target compressions were too shallow, as there was no value above target (> 6 cm) in either group (Figure 3).

Mean compressions rates were similar in both groups (116 cpm (95% CI 112–120) in the experimental group versus 115 cpm (95% CI 110–119), *p* = 0.65), with a similar proportion of within-target compressions between groups (median (Q1; Q3) 78.2% (52.9; 92.6) versus 89.9% (62.3; 96.9), *p* = 0.40) (Appendix A). There was no difference regarding chest recoil (median (Q1; Q3) 97.9% (86.7; 99.6) in the experimental group versus 98.9% (92.4; 99.5), *p* = 0.90).

Time to first shock was also similar (mean 41.6 s (95% CI 36.1–47.2) in the experimental group versus 40.9 (95% CI 34.7–47.1), *p* = 0.85). There was no difference regarding time to first ventilation (median (Q1; Q3) 109.0 s (90.3; 126.5) in the experimental group versus 102.6 (93.1; 110.1), *p* = 0.74). The number of shocks delivered was slightly higher in the experimental group (median (Q1; Q3) 5 (5; 5) versus 4 (4; 5), *p* = 0.05).

The total number of ventilations over the 10 min scenario was more than twice as high in the experimental group (mean 39 (95% CI 33–46) versus 19 (95% CI 16–23), *p* < 0.001), corresponding to a higher minute ventilation (median (Q1; Q3) 2374 mL/min (2134; 2672) versus 794 mL/min (689; 1285). The proportion of ventilations within target volume was also significantly higher in this group (median (Q1; Q3) 94.6% (88.9; 98.0) versus 81.8% (65.0; 85.7), *p* = 0.003) (Figure 4).

Mean ventilation volume was higher in the experimental group (481 mL (95% CI 430–533) versus 385 mL (95% CI 347–423), *p* = 0.003); combined with the number of ventilations, this led to a higher minute ventilation (median (Q1; Q3) i-gel^®^ 2044 mL (1709; 2185) versus 662 mL (540; 998), *p* < 0.001).

There was no difference in the self-assessed cognitive load (4.4 (95% CI 3.4–5.3) in the experimental group versus 4.7 (95% CI 3.9–5.5), *p* = 0.61). Satisfaction was higher in the experimental group (*p* = 0.01) (Appendix A).

## 4. Discussion

In this simulation study, the immediate insertion of an i-gel^®^ device led to significantly higher CCFs. This difference can be explained by the ability to immediately provide continuous chest compressions and by the avoidance of interruptions otherwise necessary to provide ventilations when using a BVM device. CCF was consistently higher during the following cycles, even during one rescuer CPR (applying 30:2 scheme). This implies that using an i-gel^®^ can help decrease no-flow time by overriding the time required to adequately place the ventilation mask. In contradiction with Vogt et al.’s study [55] but similarly to Cereceda-Sánchez et al.’s findings [56], ventilations were also of higher quality when the i-gel^®^ device was used. Indeed, the number of ventilations provided was more than twice as high in the experimental group, while hyper-insufflations remained anecdotic. These endpoints should be investigated in a pediatric population, as the experimental approach we have described could offer a substantial advantage in pediatric OHCA where ventilations are paramount [57].

The number of shocks delivered was slightly higher in the experimental group. As time to first shock and to first ventilation were similar between groups, it seems that, contrarily to the King Laryngeal Tube [58], the i-gel^®^ device does not delay the provision of these critical actions.

The present study also supports the use of a short and well-focused learning pathway, as paramedics were able to correctly insert the i-gel^®^ in all cases and without delay to critical actions. Even though a face-to-face workshop is certainly mandatory, the theoretical parts could easily be provided as asynchronous, distance learning interventions [59,60,61].

In our scenario, OHCA had to be managed by two providers only, which represents a worst-case situation. Actual clinical outcomes might prove superior to those we report as CPR-trained bystanders might be assigned to perform high-quality chest compressions while professionals focus on other aspects of OHCA-management, such as team coordination, ventilations, rhythm analysis and ALS treatments.

Compressions were unexpectedly shallower in the experimental group. Two main hypotheses could explain this fact. First, most compressions were delivered in an over-the-head position [62,63,64], and the cumbersome airway management apparatus (i-gel^®^ + antibacterial filter + BVM) coupled with a shallower i-gel^®^ insertion depth in the manikin might have prevented the adequate provision of chest compressions. Second, participants might have been too focused on the i-gel^®^ device, the use of which was new to them, thus downgrading the quality of compressions. Regardless of the actual reasons explaining these shallower compressions, this issue must be fixed before implementation of the i-gel^®^ in the field can be considered. Further studies should now be carried out to assess the most effective way of maintaining an adequate compression depth. These studies could consider two different approaches: forcing participants to deliver compressions from the patient’s side [65], using a feedback device [66,67,68,69] or both together.

### Strengths and Limitations

The main limitation is the simulated context, and clinical studies will be needed to validate this experimental approach once the issue regarding the shallow compressions has been addressed. Another limitation is that the recruitment strategy could have biased the sample through a self-selection phenomenon. Randomisation should, however, have mitigated this phenomenon.

The resuscitation sequence outlined in the demonstration video might have been different than the sequence some participants were used to. In this video, both providers initially performed CPR together and switched to single-rescuer CPR only two minutes after the first shock had been delivered to allow the other rescuer to gain intravenous access. This might have induced, in both groups, a different strategy than the one they would have chosen in their actual practice, thus affecting BLS quality. Nevertheless, randomisation should have smoothened this effect out. Moreover, it is necessary to highlight that the proposed approach may delay IV cannulation and the administration of medications. While this is hardly relevant regarding the chosen scenario of a VF, this should be assessed and, more specifically, in case of asystole or pulseless electrical activity.

Some outcomes, such as satisfaction, might be overestimated, as many participants were eager to increase their skills and knowledge and hoped to autonomously use the i-gel^®^ device in the future. In addition, a Hawthorne effect cannot be ruled out, as participants knew that they were being observed and that the results would be submitted for publication [70]. This might have happened despite the fact that participants were unaware of the study outcomes, as it was hardly possible to blind them to the type of intervention.

The strengths of this trial reside in its multicentre design, which allowed the inclusion of paramedics and EMTs from different EMS. The design was also rather original, and we believe that our results address a gap regarding the optimal approach in the initial management of OHCA, while raising other questions.

## 5. Conclusions

After minimal training, paramedics and EMTs using our experimental approach achieved a higher CCF and better ventilation parameters than those applying their standard of care. Our results show that the early insertion of an i-gel^®^ device enables these enhancements in an adult OHCA simulation without delaying time to first ventilation or to first shock. However, chest compressions were shallower when the i-gel^®^ was used, and further studies are needed to understand and address this issue before clinical trials can be considered.

## Figures and Tables

**Figure 1 jcm-11-00217-f001:**
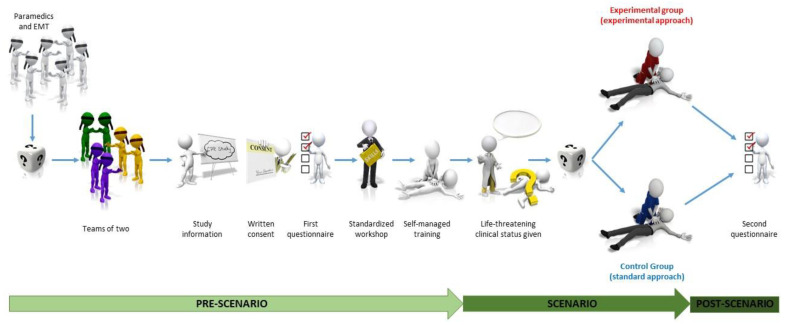
Study sequence.

**Figure 2 jcm-11-00217-f002:**
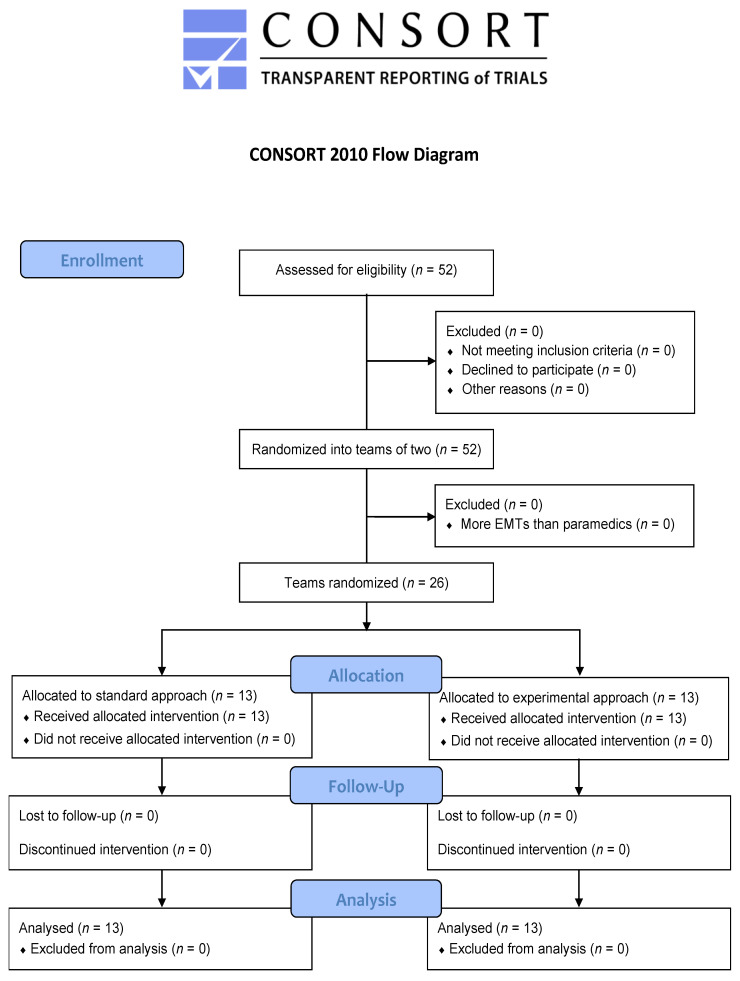
Study flowchart.

**Figure 3 jcm-11-00217-f003:**
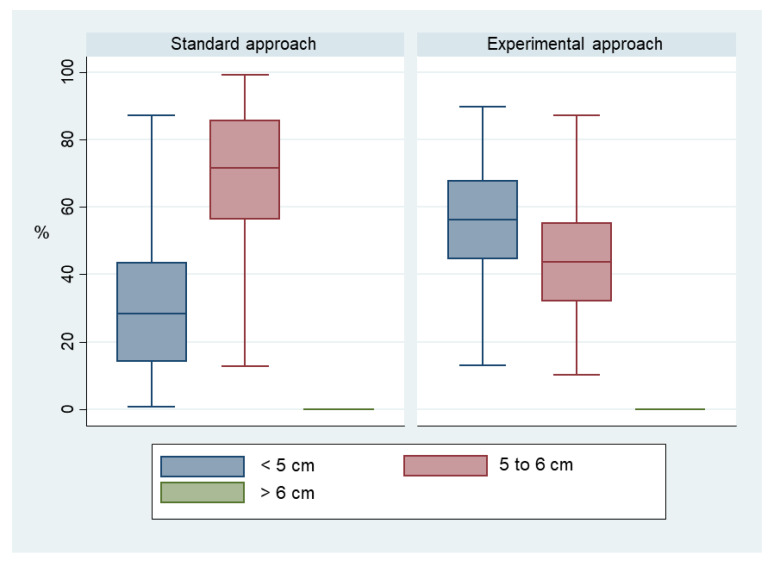
Proportions of compressions below, within and above depth target value.

**Figure 4 jcm-11-00217-f004:**
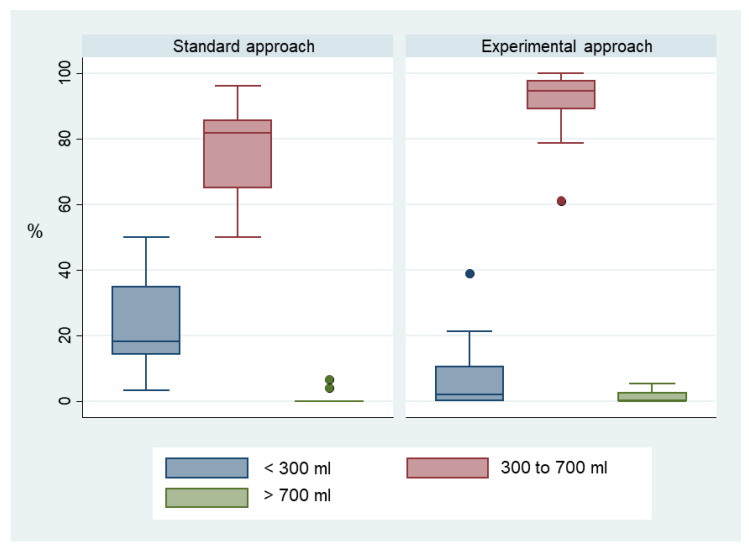
Proportions of ventilations below, within and above volume target value.

**Table 1 jcm-11-00217-t001:** Participants’ characteristics.

	Standard Approach (*n*= 26 Participants)	Experimental Approach (*n* = 26 Participants)
Age, mean (SD), years	36.6 (9.0)	35.9 (8.0)
Gender, *n* (%)		
- female	11 (42.3)	8 (30.8)
- male	15 (57.7)	18 (69.2)
- other	0 (0.0)	0 (0.0)
Experience, mean (SD), years	10.3 (8.5)	10.6 (8.3)
Function—Paramedic, *n* (%)	23 (88.5)	21 (80.8)
Number of i-gel^®^ insertions on manikin in the past year, median (Q1; Q3)	0 (0; 1)	0 (0; 2)
Number of insertions on human in the past year, median (Q1; Q3)	0 (0; 0)	0 (0; 0)

**Table 2 jcm-11-00217-t002:** Chest compression fraction, overall and per cycle expressed as mean (95% CI).

Minutes	Standard Approach (*n* = 13 Teams)	Experimental Approach (*n* = 13 Teams)	*p*-Value
0–2	83.6% (81.2–86.1)	89.0% (87.0–91.1)	0.001
2–4	67.4% (64.6–70.2)	79.0% (74.0–84.0)	<0.001
4–6	63.7% (59.9–67.6)	72.0% (67.4–76.6)	<0.001
6–8	60.1% (57.2–63.0)	70.9% (66.2–75.6)	<0.001
8–10	64.4% (59.5–69.3)	71.2% (67.1–75.4)	0.03
Overall	67.8% (65.7–70.0)	76.4% (73.1–79.7)	<0.001

## Data Availability

The datasets generated and analysed during the current study are available in Mendeley Data repository, (https://data.mendeley.com/datasets/98rf9psgvb/1) (accessed on 30 November 2021).

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
