# Peer review of "Effect of Early Supraglottic Airway Device Insertion on Chest Compression Fraction during Simulated Out-of-Hospital Cardiac Arrest: Randomised Controlled Trial"

_jcm, 2021, doi:10.3390/jcm11010217_

Round 1

Reviewer 1 Report

Abstract

Page 1 line 39: It is not clear to which group the following sentence refers to. In this group, ventilation parameters were enhanced, but compressions were significantly shallower (4.6 cm vs 5.2 cm, p=0.007). 

Methods: 

The fact that chest compressions were provided with the rescuer placed over the head of the patient should be clearly mentionned in the method section.

I'm not sure that this study can be called a multicenter study as, if I understood correctly, all the simulation and assessments were done in the same simulation center. The fact that participants are recruited from different ambulance services is to me insufficient to call the study a multicenter study.

Results:

Page 8 line 240: The following sentence needs rewording as it is too long and it is not clear which number refers to which group. Time to first shock was also similar between groups (mean 41.6 seconds [95%CI 36.1 - 47.2] in the experimental group versus 40.9 [95%CI 34.7 - 47.1], p=0.85), and there was no statistically significant difference regarding time to first ventilation (median [Q1; Q3] 109.0 seconds [90.3; 126.5] versus 102.6 [93.1; 110.1], p=0.74).

Page 8 line 245: It should be mentioned the unit of time during which the total number of ventilation were measured. The authors should consider to present the number of ventilations per minute in order to allow the reader to better appreciate on the appropriateness of the ventilation rate. The total number of ventilations was more than twice as high in the experimental group (mean 39 [95%CI 33 - 46] versus 19 [95%CI 16 - 23], p<0.001).

Discussion:

Page 8 line 259: The authors should consider to suppress the word 'significantly'

Page 9 line 272: The authors might consider rewording the following sentence without overstressing the statistical significance:The number of shocks delivered was slightly higher in the experimental group, but 
this difference failed to reach statistical significance.

It should be mentionned in the discussion that the proposed method delays IV canulation and the administration of medication which is irrelevant in the chosen scenario of a VF but can affect the outcome of the patient in a scenario with an asystole or a pulseless electrical activity.

Conclusions:

The authors might consider that the conclusion is an overstatement when saying After minimal training, paramedics and EMTs using our experimental approach per formed better than those applying their standard of care.

Some aspects of the resuscitation were better (notably CCF) and some worse (chest compression depth). The conclusion should be reworded to acknowledge those specific results.

Reviewer 2 Report

Congratulations

It is well prepared randomized simulation study of correlation between SGA use during OHCA CPR. The results are well prepared and presented. Authors proved that early insertion of a SGA device may improve CCF. Authors included limitations which are very important in simulation study, cause clinical validation should be performed.

I have no additional concerns. Article is well prepared and well written. Methodology is sufficient and results presentation is correct. Discussion and conclusions are proper and fulfill the subject.
